# OpenReview forum: "Don't Walk the Line: Boundary Guidance for Filtered Generation"
_ICLR.cc/2026/Conference — Submitted to ICLR 2026_

### Official Review · Reviewer_ekjX · 2025-10-30

**Soundness:** 1
**Presentation:** 1
**Contribution:** 2
**Rating:** 2
**Confidence:** 4

**Summary:**

This paper proposes Boundary Guidance (BG), a reinforcement-learning-based fine-tuning approach for generative models paired with downstream safety classifiers. The core idea is to explicitly steer generations away from the decision boundary of the classifier, rather than minimizing rejection probability only. By doing so, the approach aims to reduce both false positives and false negatives in filtered generation systems.

The authors present a decision-theoretic framework that formalizes how utility is minimized near classifier boundaries and derive a boundary-avoiding reward function combining helpfulness and safety signals. Empirical results across multiple model families (Qwen2.5 and Gemma) and sizes (0.5B–14B) demonstrate consistent improvements in both helpfulness and harmlessness as evaluated by GPT-4.1.

**Strengths:**

This paper focuses on improving the safety of generative models, which is an important topic in this field. The high-level idea is easy to grasp and understand.

**Weaknesses:**

1. [Quality] Limited baselines: the experiments only compare with the base model, which is far from sufficient for validating the effectiveness of the methods.

2. [Quality] Too heuristic: the design of the algorithm is too heuristic, which lacks both theoretical justifications and empirical validation via ablation studies.

3. [Clarity] Figure 2, 3: the naming of models are inconsistent, e.g. "qwen25-0.5b-guard_only", "Qwen2.5-0.5B-instruct", which poses difficulties for readers to quickly grasp the point of the figures.

4. [Clarity] Messy organization: Section 5 describes the main setup of the experiment, but is titled "Experimental Evidence for Boundary Guidance".

**Questions:**

* lines 169-171, Equation 2: If $t(x,y)$ is the probability of unsafe outputs (lines 160–161), and the goal is to minimize likelihood, I think the utility should be defined as $-\log t(x,y)$, not $-t(x,y)$, since the ultimate probability is the product of each event instead of the sum, if we consider different samples to be independent from each other, as in the conventional Maximum Likelihood Estimation (MLE) paradigm.

---

> ### Author Response · Authors · 2025-11-21
>
> Thank you for your thorough review! We respond to your points in turn.
> 1. We compare to the base model plus a guard model, and clarify this in the revised manuscript.
> 2. We are not aware of methods that are directly comparable to us in that they: a) condition on a classifier input b) don’t require resampling inference time (e.g., rejection sampling) and c) are full-sequence in that they use classification probability at the end of generation.
> 3. In the original paper, we consider multiple ablation studies with multiple model sizes and three different reward specifications. We add in our revised manuscript different classifiers, and different LLM judges. We also add a derivation of our algorithm relying on large-margin classification’s reduced complexity for statistical learning. If you would like us to do additional ablations to clear the bar, please let us know. Note that reviewers pKz2, 2aTR highlight that we do extensive ablations.
> 4. “Qwen2.5-Instruct” refers to the filtered base model, Qwen2.5-Guard-Only” is finetuned using a reward R’. We made the naming consistent in the revised paper.
> 5. We rename Section 5 to “Experimental Setup” in the revised manuscript.
> 6. Our model does not consider independent draws of the safety classifier, as it is run once at the end of generation (either after a maximal number of tokens or when the eos token is sampled). The relevant expected system utility is indeed -t(x,y).

---

### Official Review · Reviewer_2aTR · 2025-10-31

**Soundness:** 3
**Presentation:** 2
**Contribution:** 3
**Rating:** 6
**Confidence:** 3

**Summary:**

The authors propose a new method, Boundary Guidance, that improves existing RL-techniques for fine-tuning models towards safer outputs. Existing methods might lead to examples close to the classifier's margin, which their method discourages. They carry out extensive experiments and ablations to demonstrate the utility and robustness of their approach.

**Strengths:**

- clear motivation and problem framing: as generators are trained independently of downstream safety filters, that is causing outputs near the classifier boundary
- simple and effective idea (introducing a penalty for being close to the classifiers boundary)
- strong empirical results that show Pareto improvements
- thorough ablations

**Weaknesses:**

- evaluation depends heavily on LLM-as-a-judge --> human ablation would be useful
- assumes that classifier probabilities near the extremes are reliable and that the decision boundary is meaningful
- limited dataset diversity (esp. no long-context scenarios)

**Questions:**

- How does Boundary Guidance perform under adaptive adversarial attacks where the attacker optimizes specifically to evade the classifier confidence signal?
- How sensitive is the performance to the specific safety classifier used? What about a "worse" classifier?
- Have you conducted any human evaluations to validate the LLM-as-a-judge setup?

---

> ### Author Response · Authors · 2025-11-21
>
> Thank you for your thorough review. We respond to each of your points:
> 1. We test against Gemini as an additional LLM judge, finding without parameter tuning significant Pareto improvement for the largest tested model Qwen2.5-14B. We also add a human ablation in our next pdf revision.
> 2. We make our assumption on the reliability of extreme predictions explicit, and add a discussion of the simplicity of learning large-margin classifiers.
> 3. We add long-context prompts in a novel set of experiments, finding Pareto improvements.
> 4. We test with other classifiers in the revised manuscript. We find that the method requires a safety classifier that is not much weaker than the base model to work.

---

> > ### Comment · Reviewer_2aTR · 2025-11-27
> >
> > Thanks for clarifying the assumptions and adding experiments. At this moment, I will maintain my score, but I am open to reconsider if the manuscript is updated.

---

### Official Review · Reviewer_pKz2 · 2025-11-01

**Soundness:** 3
**Presentation:** 3
**Contribution:** 2
**Rating:** 4
**Confidence:** 3

**Summary:**

This paper proposes Boundary Guidance, a reinforcement learning fine-tuning method for improving compound safety systems where generative models are paired with downstream safety classifiers. The method stems from the insight that standard fine-tuning approaches often push models to generate outputs near classifier decision boundaries, increasing false positives and false negatives. Boundary Guidance steers generation away from these decision boundaries using a reward function that combines utility signals from reward models with safety classifier confidence scores. Experimental results show improvements in safety and utility across multiple model scales on jailbreak and ambiguous prompt benchmarks.

**Strengths:**

- The method focuses on compound systems rather than isolated model training, which aims to address the gap in current safety studies.

- The results demonstrate consistent Pareto improvements across multiple model architectures and scales. The experiments include various benchmarks and an LLM-as-a-Judge evaluation. The ablation experiments help understand the method components.

**Weaknesses:**

- The method makes strong simplifying assumptions, such as the constant utility $u(x,y)$ and binary safety classification, which may not hold in real-world scenarios. The analysis also assumes perfect classifier calibration and does not discuss how the approach performs when classifiers are biased or when the decision boundary itself is poorly positioned.

- The evaluation uses GPT-4.1 as the sole judge for both helpfulness and harmfulness, which can introduce potential evaluation bias, especially when fine-tuning against other LLM-based safety classifiers. The evaluation is also limited to a single benchmark focused on jailbreaks and harmful prompts. Many of the improvements are relatively small.

- The approach requires training with three models, which may be computationally prohibitive.

**Questions:**

- How does the method perform when the classifiers are miscalibrated?

---

> ### Author Response · Authors · 2025-11-21
>
> We thank the reviewer for their thorough feedback!
> 1. We first want to point out that our decision-theoretic model makes strong assumptions to justify the method, but that our method does not need to make these assumptions. In fact, we deploy it with non-constant utility and non-calibrated safety classifiers. We show that our method yields Pareto improvements even in the absence of our assumptions. Hence, our theoretical model should be seen as a reason for why intuitively boundary guidance should perform well. Still, to remedy the limitations of the model, we add a discussion based on the lower complexity of large-margin classification compared to lower-margin classification, which does not rely on simplifying assumptions. We also discuss how to relax the restriction to binary safety classifiers in our discussion section.
> 2. We add long-context prompts to increase the diversity of evaluation and training and discuss it in our revised manuscript. We find a Pareto improvement for these prompts as well.
> 3. We also evaluate our model against Gemini, yielding Pareto improvements for the largest model without any hyperparameter tuning (see revised manuscript).
> 4. In fact, we show that equally good results (with the exception of Qwen2.5-0.5B) are obtained with the guard-only model, which only trains to avoid the decision-boundary. In this case, only two models are needed.
> 5. Empirically, safety classifiers are, like other deep classifiers, most likely to be miscalibrated [1]. Our discussion of the reduced complexity of large-margin generation relaxes the restriction on calibration for our theoretical results.
>
> [1] Guo, C., Pleiss, G., Sun, Y., & Weinberger, K. Q. (2017). On calibration of modern neural networks. Proceedings of the 34th International Conference on Machine Learning (ICML), 1321–1330. PMLR. https://arxiv.org/abs/1706.04599

---

### Official Review · Reviewer_sPq8 · 2025-11-06

**Soundness:** 2
**Presentation:** 2
**Contribution:** 2
**Rating:** 2
**Confidence:** 4

**Summary:**

The paper addresses the challenge of balancing safety and utility in the design of algorithms for safe compound systems. With a simple utility-safety model, the paper argues that the utility tends to diminish near the safety-only decision boundary, which constrains helpfulness and motivates incorporating utility directly into the reward formulation. To avoid explicitly estimating the trade-off coefficient (the $\lambda$ term) between safety and utility, motivated by the fact that models with low confident safety assessment tend to perform poorly, the paper proposes a reward function that includes a direct utility component, while safety is enforced through a boundary-maximizing objective that encourages the model to be confident about the safety evaluation (away from the threshold of 0.5, which represents uncertainty in safety evaluation). Empirical results from fine-tuning models of various scales show consistent improvements in helpfulness while simultaneously reducing harmfulness, validating the effectiveness of the approach.

**Strengths:**

1. The question of how to bring the aspect of utility into the safety aware compound systems is interesting.
2. The formulation in Section 3 is good and well-written.

**Weaknesses:**

1. The boundary objective in Eq (3) lacks detailed explanation and motivation. Consider including a discussion paragraph on how it is derived/proposed.

2. The connection of Eq (3) to Eq (2) and the previous discussion almost seems unrelated. The final objective is basically a utility maximization with a boundary-maximizing objective for safety. It is critical to explain the connection of Eq (3) to Section 3.

3. It seems that there is no way to fully decouple the effect of $u(x,y)$ and $t(x, y)$, which makes the algorithm work well, i.e., $u(x, y)$ (utility/helpfulness) still captures some effect of $t(x, y)$ (safety/harmlessness). Otherwise, why is there a utility maximization term when the output is blocked, i.e., when $t(x,y) \ge 0.5$. This is contradictory to the Eq (2) and footnote 1.

4. No empirical comparison with prior work.

**Questions:**

Questions:

1. Can you please confirm whether the paragraph after Proposition 1 is comparing the aggregate expected reward in a scenario with an individual utility function, with the one without any individual utility function? If so, please provide the exact expected reward similar to Eq (1), when there is no utility function $u(x,y)$. Coming back to the question: why is having a **local** minima at $\tau$ important?

2. Can you confirm that Base refers to the evaluation of responses by the base model? If so, please compare these with Base+Guard (filtering with $t(x,y)$), which does not require training.

Typos:
1. Line 045: the approach $\rightarrow$ these approaches.
2. Figure 2: Qwen25 $\rightarrow$ Qwen2.5,

---

> ### Author Response · Authors · 2025-11-21
>
> Thank you for your thorough feedback!
> 1. We first want to highlight that we do not formulate our objective in a way that increases utility as part of the objective. In fact, our theoretical analysis in Section 3, in particular (2), highlights that even if all generations yield the same utility, we derive higher system utility as fewer over-blocked and under-blocked completions are generated. Also, our empirical results show that the utility model is not necessary to achieve a Pareto improvement of harmlessness and helpfulness, see Table 2.
> 2. We include an additional justification of the reward (3) in our revised manuscript. We hope that this addresses weaknesses 1-3.
> 3. Proposition 1 considers a special case of constant utility to the individual, but reasons about the system utility. The result shows that system utility has a *global* minimum at the decision boundary tau. We make this more explicit in the revised manuscript.
> 4. Our guard evaluation is against the base model including a guard.
> 5. Typos: Fixed. Thank you!

---

### Author Response · Authors · 2025-11-21

We thank the reviewers for their feedback on our paper. We are grateful that reviewers found our question “interesting” (Reviewer sPq8), found our theoretical formulation “good and well-written” (Reviewer sPq8), our experiments “extensive” (2aTR), and our approach “robust” (2aTR). The main concerns in the reviews are (1) our restriction to one LLM-as-a-Judge, (2) our restricted set of prompts, (3) our assumptions in our theoretical analysis (4) our limited set of tested classifiers. We address all four of them, in addition to the questions of the reviewers in our revisions, and hope that we convince the reviewers that Boundary Guidance is simple, portable, and robust enough to warrant publication in the ICLR proceedings.

We also add an algorithm pseudocode and our replication package/code in the manuscript.

---

### Author Response · Authors · 2025-12-03

Dear AC,
welcome to this paper. We point out that despite the scores which are at times low, the reviewers state that they found our question “interesting” (Reviewer sPq8), our theoretical formulation “good and well-written” (Reviewer sPq8), our experiments “extensive” (2aTR), and our approach “robust” (2aTR). The main concerns in the reviews were (1) our restriction to one LLM-as-a-Judge (we addressed this in a revision), (2) our restricted set of prompts (we addressed this in a revision), (3) our assumptions in our theoretical analysis (we addressed this in a revision) (4) our limited set of tested classifiers (we addressed this in a revision). We list our changes to address the concerns in responses to the individual reviewers. We hope that this is sufficient evidence to show you that Boundary Guidance is simple, portable, and robust method deserving publication in the ICLR proceedings.

---

### Meta-Review · Area_Chair_Ykpz · 2026-01-08

**Summary:**

The original primary concerns of the reviewers are: the design of the algorithm is very heuristic-y (with Eq 3's derivation being a particular point of contention), limited baselines, and lack of clarity in positioning with respect to baselines. That said, the reviewers do find the overall premise of the paper interesting and seem to think it is worth the research effort. On the whole, the initial evaluation indicates that the weaknesses outweigh the strengths of the work.

**Reviewer Concerns:**

The not enough baselines-esque arguments are mostly addressed by the reviewers, though they could do a much better job of explaining exactly what the changes were instead of expecting reviewers to go line by line in the updated draft. There are now multiple LLM as judges, multiple types of guard models, and more robust long context evaluations in the main paper itself.

That said, the things that are not quite so well addressed include the justifications of the Eq 3 etc. - subsections on decision and learning theory justifications have been made but seem rather cursory and do not give the reader a good sense for why these frameworks are justified in the first place.

**Reviewer Scores:**

sPq8: would have (with low probability) increased the score to a 4 but likely no further given their justification concerns

pKz2: had many of their concerns addressed so they possibly could have increased to a 6

2aTR: says they will maintain their score after reading the rebuttal

ekjX: would have increased their score to 4 with a (higher probability) than sPq8, given most of their concerns were about lack of sufficient baselines and less about the theoretical justifications

---

### Decision · Program_Chairs · 2026-01-26

Reject